# *Schistosoma japonicum* cathepsin B2 (SjCB2) facilitates parasite invasion through the skin

Bingkuan Zhu[1], Fang Luo[1], Yi Shen[1], Wenbin Yang[1], Chengsong Sun[1], Jipeng Wang[1], Jian Li[2], Xiaojin Mo[3], Bin Xu[3], Xumin Zhang[1], Yongdong Li[4]\*, Wei Hu[1,3]\*

**1** Department of infectious diseases, Huashan Hospital, State Key Laboratory of Genetic Engineering, Ministry of Education Key Laboratory for Biodiversity Science and Ecological Engineering, Ministry of Education Key Laboratory of Contemporary Anthropology, School of Life Science, Fudan University, Shanghai, China, **2** Dermatology Department, Huashan Hospital of Fudan University, Shanghai, China, **3** National Institute of Parasitic Diseases, Chinese Center for Disease Control and Prevention, Key Laboratory of Parasite and Vector Biology of China Ministry of Health, WHO Collaborating Centre for Tropical Diseases, Joint Research Laboratory of Genetics and Ecology on Parasite-host Interaction, Chinese Center for Disease Control and Prevention & Fudan University, Shanghai, China, **4** Key Laboratory of Organo-Pharmaceutical Chemistry, Gannan Normal University, Ganzhou, China

\* 172123135@qq.com (YL); huw@fudan.edu.cn (WH)

**Data Availability Statement:** All relevant data are within the manuscript and its Supporting Information files.

## Abstract

Cercariae invasion of the human skin is the first step in schistosome infection. Proteases play key roles in this process. However, little is known about the related hydrolytic enzymes in *Schistosoma japonicum*. Here, we investigated the biochemical features, tissue distribution and biological roles of a cathepsin B cysteine protease, SjCB2, in the invasion process of *S. japonicum* cercariae. Enzyme activity analysis revealed that recombinant SjCB2 is a typical cysteine protease with optimum temperature and pH for activity at 37°C and 4.0, respectively, and can be totally inhibited by the cysteine protease inhibitor E-64. Immunoblotting showed that both the zymogen (50 kDa) and mature enzyme (30.5 kDa) forms of SjCB2 are expressed in the cercariae. It was observed that SjCB2 localized predominantly in the acetabular glands and their ducts of cercariae, suggesting that the protease could be released during the invasion process. The protease degraded collagen, elastin, keratin, fibronectin, immunoglobulin (A, G and M) and complement C3, protein components of the dermis and immune system. In addition, proteomic analysis demonstrated that SjCB2 can degrade the human epidermis. Furthermore, it was showed that anti-rSjCB2 IgG significantly reduced (22.94%) the ability of the cercariae to invade the skin. The cysteine protease, SjCB2, located in the acetabular glands and their ducts of *S. japonicum* cercariae. We propose that SjCB2 facilitates skin invasion by degrading the major proteins of the epidermis and dermis. However, this cysteine protease may play additional roles in host-parasite interaction by degrading immunoglobins and complement protein.

## Author summary

Schistosomiasis is one of the most prevalent parasitic diseases in the world, with about 200 million humans infected in 74 tropical countries. The infection of schistosome is initiated when the larvae, cercariae, penetrate the human skin. Proteolytic enzymes are likely

**Funding:** The project was funded by the National Key research and Development Project of China (Grant No.: 2018YFA0507300) and the National Natural Science Foundation of China (Grant No.: 31572513) to Wei Hu. The website of the funder "Ministry of Science and Technology of the People's Republic of China" is http://www.most.gov.cn/. The website of the funder "the National Natural Science Foundation of China (NSFC)" is https://isisn.nsfc.gov.cn. The funders had no role in study design, data collection and analysis, decision to publish, or preparation of the manuscript.

**Competing interests:** The authors have declared that no competing interests exist.

involved in the invasion process, but these have yet to be characterized for *S. japonicum*. Here, we have functionally expressed a recombinant form of the cathepsin B cysteine protease SjCB2 in the yeast *Pichia pastoris*. Our study showed that SjCB2 degraded a number of proteins associated with the skin and immune systems, and disrupted the structure of the human epidermis. The enzyme was located in the acetabular glands and their ducts in the cercariae, where it would be stored before released into the skin. Antibody-blocking studies revealed that SjCB2 had a 22.94% contribution during the cercariae invasion process. Taken together, our findings suggest that SjCB2 helped cercariae penetrating the skin barrier and evading the immune attack to allow successful infection in the mammalian host.

## Introduction

Schistosomiasis, also known as bilharzia or snail fever, is a chronic infectious disease caused by trematode blood flukes of the genus *Schistosoma*. These parasitic worms infect over 200 million people in 74 tropical and subtropical countries [1, 2], and is categorized as a global neglected tropical disease (NTD) [3]. The three major species of schistosome that infect humans are *Schistosoma haematobium*, which causes urinary schistosomiasis [4], *Schistosoma mansoni* and *Schistosoma japonicum* that both cause intestinal schistosomiasis. Adult schistosome worms reside in the veins as male/female pairs producing hundreds to thousands of eggs per day [5]. Eggs can become entrapped in intestines or liver (for *S. mansoni* and *S. japonicum*) or bladder and urogenital system (for *S. haematobium*), and induce granulomatous reactions that are regarded as the main pathogenic factor of schistosomiasis [6]. Praziquantel is the only drug used in most schistosomiasis treatment, while recent studies suggest that resistance to this drug may be emerging [5]. Therefore, the development of vaccines as well as new anti-schistosome drugs are urgently needed [7].

Human become infected with schistosome parasite if cercariae released by the infected intermediate snail host penetrate the skin and enter the bloodstream. Schistosome cercariae find their host by using different cues such as motion, light-dark contrast, and chemical and thermal gradients [8–10]. Carbon-free fatty acids released from the skin, such as linoleic acid, can attract the cercariae [11, 12]. After reaching the skin, the cercariae crawl along the epidermis to find a suitable position to attach using ventral suckers [13]. Mammalian skin is a formidable barrier formed by epidermis and dermis [14]. When the invasion process begins, mucins are released from the post-acetabular glands to allow adherence to the host skin while proteolytic enzymes released from the pre-acetabular glands degrade skin proteins [15, 16]. Several studies have characterized cercarial enzymes involved in the penetration process, the most well-known being the *S. mansoni* S1A serine protease, and cercarial elastase (SmCE) [17–19]. Because *S. japonicum* exhibits a more rapid migration through the skin compared with *S. mansoni* and *S. haematobium* [20], it is speculated that *S. japonicum* may use a different penetration mechanism [21]. A comparative study showed that the acetabular glands contents of *S. japonicum* cercariae had a 40-fold greater cathepsin B (CB) cysteine protease-like activity compared to *S. mansoni* cercariae, suggesting that CB played an important role in *S. japonicum* cercariae invasion [22]. Our previous proteomic study revealed that SjCB2 and SjCE2b were consumed in the skin penetration process, inferring that these two enzymes may be both involved in the cercariae invasion [23]. However, it is not clear what role SjCB2 plays in the invasion process of *S. japonicum* cercariae.

In this study, we expressed a functionally active recombinant SjCB2 in the methylotrophic yeast *Pichia pastoris* and characterized the biochemical properties of the purified enzyme. The *in vitro* degradation assay showed that several proteins of human dermis, immune system and

blood circulatory system could be digested by this enzyme. Stable isotope dimethyl labeling proteomic analysis revealed the destruction effect of SjCB2 on the cultivated human epidermis. Immunofluorescence localized SjCB2 in the cercariae acetabular glands and their ducts. In the end, we showed that anti-rSjCB2 IgG reduced the cercariae invasion ability. Our results indicated that SjCB2 was involved in the *S. japonicum* cercariae invasion process, and also offered a new target for novel anti-infection therapeutics.

## Materials and methods

### Ethics statement

All experiments involving animals were carried out in accordance with the guidelines for the Care and Use of Laboratory Animals of the Ministry of Science and Technology of the People's Republic of China (2006398) and approved by the Ethics and Animal Welfare Committee of the National Institute of Parasitic Diseases, Chinese Center for Disease Control and Prevention, Shanghai, China (IPD2008-4).

### Parasites and animals

A Chinese mainland isolate of *S. japonicum* from Anhui province, and the infected *Oncomelania hupensis* snails were provided by the pathogen biology laboratory of the National Institute of Parasitic Diseases, Chinese Center for Diseases Control and Prevention, Shanghai. Six-week-old female C57BL/6 strain mice were purchased from Shanghai Animal Center, Chinese Academy of Sciences (Shanghai, China).

### Bioinformatics data mining and molecular modeling of SjCB2

Sequence of SjCB2 was downloaded from NCBI with the accession number AY226984.2. BLASTx (http://www.ncbi.nlm.gov/BLAST) was used to analyze sequence identity and positivity. Protein domains were analyzed by SMART tool (http://smart.embl-heidelberg.de/smart/set_mode.cgi?NORMAL=1). Multiple sequence alignments were carried out by ESPript 3.0 server (http://espript.ibcp.fr/ESPript/cgi-bin/ESPript.cgi) using CBs from *Schistosoma mansoni*, *Schistosoma haemotobium*, *Trichobilharzia regenti*, *Trichobiharzia szidati*, *Clonorchis sinensis*, *Opisthorchis viverini*, *Eudiplozoon nipponicum*, *Hymenolepis microstoma*, *Echinococus multiocularis*, *Macrotomum lignano*, *Angiostrongylus carvntonesis* and *Caenorhabditis elegans*. A phylogenetic tree was constructed by Maximum Likelihood method and decorated by the iTOL server (https://itol.embl.de/).

A spatial model of SjCB2 was constructed by homology modeling as described previously [24]. Briefly, The X-ray structure of *S. mansoni* cathepsin B1 zymogen (PDB code: 4I04) was used as the template. The homology module was generated by MODELLER using the Basic Modeling method. Then the model was refined by Chiron (https://dokhlab.med.psu.edu/chiron/login.php) and evaluated by the SAVES v5.0 (http://servicesn.mbi.ucla.edu/SAVES/). The docking between SjCB2 and inhibitor E-64 was implemented by the AutoDock 4.2 software. Molecular images were generated with Pymol software.

### Expression and purification of recombinant SjCB2 in *E. coli*

The open reading frame (ORF) of SjCB2 was amplified with forward primer 5'-CGGGATCC ATGTATTGGTACAATTATTATTTATTACTATGT-3' and reverse primer 5'-CCCTCGAG CTATTTTTTAATTTTCGGTATTCCA-3' using PCR Master Mix (Takara Bio, Shiga, Japan) and cDNA template prepared from adult worms. The PCR product was digested with *BamH*I and *Xho*I (NEB, USA) enzymes and gel-purified, then cloned into expression plasmid pGEX-

h-6t-1. Positive clones were screened and confirmed by DNA sequencing. The restructured plasmid was transformed into *E. coli* BL21 (DE3) (Takara Bio, Shiga, Japan), and the transformed cells were grown by shaking (220 rpm) in LB medium containing 100 μg/ml ampicillin. rSjCB2 was expressed as a glutathione-S-transferase-SjCB2 fusion protein induced by 1.0 mM isopropy1-β-d-thiogalactoside (IPTG) at 37˚C for 6 h. After induction, the cells were harvested by centrifugation at 8,000 g for 30 min at 4˚C. The inclusion bodies were purified by Ni$^{2+}$-NTA affinity-chromatography (GE healthcare, USA). We used the Amicon Ultra-10K device to concentrate the target protein, and at the same time, the elution buffer was exchanged with resolving buffer (50mM Tris, 8M urea, 300mM NaCl, pH = 8.0). Then the protein concentration was determined by a Bradford Protein Assay kit (Glory, USA).

## Production of antibody to recombinant SjCB2

Antiserum was raised in two New Zealand white rabbits by Shanghai Youke Biotechnology Co., Ltd. The first injection was administered intraperitoneally in Freund's Complete Adjuvant on the first day, whereas Freund's in-complement Adjuvant was used for the second and third time on days 15 and 43, respectively. The control group rabbit was immunized with PBS. The serum was collected at day 53. IgG was isolated from the collected serum using Protein G-Sepharose (1 ml) as described [25] and stored at -20˚C. Because there is a GST tag in the recombinant SjCB2, we removed the anti-GST IgG by incubating the antibody with purified GST protein at 4˚C overnight. The next day, the supernatant without GST IgG was collected by centrifugation at 5,000g for 10 min at 4˚C and stored at -20˚C.

In order to analyze the immune response to rSjCB2, IgG titers of the two rabbits' serum against rSjCB2 at day 53 post injection were determined by enzyme-linked immunosorbent assay (ELISA). Briefly, the microtiter plates were coated with 2 μg/ml purified rSjCB2 and incubated at 4˚C overnight. The plates were blocked with 5% skim milk for 2 h at 37˚C. Then the plates were washed three times with PBS-T. After washing, the plates were incubated with serial dilutions of the immune serum from rSjCB2 protein for another 2 h. HRP-conjugated Goat anti Rabbit IgG (1:8,000 dilutions) (BBI, Shanghai, China) were used as the secondary antibody. Finally, 100 μl TMB (TIANGEN BIOTECH, Beijing, China) was added to the plate, the absorbance was measured at 450 nm after adding 2 M $H_2SO_4$ to stop the reaction.

In order to analyze the purity of the collected IgG, IgG titers against GST of the anti-GST-rSjCB2 IgG and the purified anti-rSjCB2 IgG were determined by ELISA. The reactions were developed using TMB as described above.

## Preparation of cercariae extracts and immunoblotting

To prepare the cercariae soluble protein extracts, dechlorinated water containing about 30,000 cercariae was placed on ice for 1 h, then centrifuged for 30 min at 4,000 g, 4˚C. The supernatant was discarded, then the cercariae deposit was homogenized by sonication (40% power, 5s on, 10s off, 10 min, Ultrasonic processor, ULTRASONIC) on ice in PBS. The extracts were centrifuged for 15 min at 13,000 g, 4˚C, ultra-filtered using a 0.22 μm Ultra free-MC device (Millipore Corporation, USA) and stored at -80˚C.

For immunoblotting, cercariae extracts (50 μg protein) was resolved by SDS-PAGE in 12% gel and electroblotted onto a PVDF membrane. The membrane was blocked in 5% non-fat dry milk in TBS-T (100mM Tris-HCl, 100mM NaCl, pH 8.0, 0.05% Tween 20) for 2 h at 37˚C, then washed with TBS-T and incubated overnight with anti-rSjCB2 IgG diluted 1:1000 in TBS-T at 4˚C. After washing with TBS-T, membranes were incubated for 2 h at room temperature in blocking solution containing HRP-conjugated Goat anti Rabbit IgG (BBI, Shanghai,

China) at a dilution of 1:5000. After washing in TBS-T, the membrane was developed with NcmECL Ultra solution (NCM Biotech, Suzhou, China) and imaged using Tanon 5200 (Tanon, Shanghai, China).

## Immunofluorescence microscopy

For immunofluorescence analysis, a protocol adapted from Ishida *et al* and Collins *et al* was used [26, 27]. Briefly, cercariae were fixed in 4% paraformaldehyde for 25 min at room temperature, then rinsed in PBSTw (PBS + 0.1% Tween-20). Next, samples were digested in permeabilization solution (2μg/ml Protease K, 0.5% SDS in 1×PBSTw) at 37°C for 40 min, and postfixed in 4% paraformaldehyde for 10 min at room temperature. After permeabilization, samples were rinsed by PBSTw, then incubated with blocking solution (5% horse serum, 0.45% fish gelatin, 0.3% Triton X-100, 0.05% Tween 20 in 1×PBS) at room temperature for 4 h. After blocking, samples were incubated with anti-rSjCB2 IgG in blocking solution with a concentration of 1 μg/ml at 4°C overnight. After > 6 h washing, samples were incubated in goat anti-rabbit Alexa Fluor 488 (Abcam, Shanghai, China) in blocking solution at 4°C overnight. Then samples were washed with PBSTw (> 6 h) containing DAPI (1μg/ml) (Abcam, Shanghai, China). After washing, samples were mounted onto a glass slide using Antifade Mounting Medium (YEASEN, Shanghai, China) to prevent the fading of fluorescence during microscopic examination. Visualization was performed on a Nikon positive confocal laser-scanning microscope using 40× immersion objective. There were two controls, the pre-immune IgG (1μg/ml) control and no primary control. The no primary control samples were incubated in blocking solution during the primary incubation step.

## Expression and purification of recombinant SjCB2 in *P. pastoris*

The ORF of SjCB2 appending a 6×His tag on the C-terminal was synthesized by GenScript Corporation and inserted into the *Xba*I-*EcoR*I (NEB, USA) double digested expression vector pPICZαA. The construct was transformed into DH5α and positive clones were selected and confirmed by DNA sequencing.

A protocol was adopted using wide type strain X33 of the methylotrophic *P. Pastoris* (Invitrogen) as a host strain [25]. Plasmid SjCB2-pPICZαA was linearized with *Sac*I (NEB, USA) and transformed into *P. pastoris* strain X33 by electroporation at 2kV, 25μF, 200Ω in electroporation cuvettes (2mm gap, Bio-Rad). The transformants were cultured on YPDS plates containing 100 μg/ml Zeocin at 28°C for 3 days to screen for colonies that have the transformed gene integrated into the yeast chromosomal DNA. A single colony of X-33-SjCB2-pPICZαA was inoculated in 10 ml BMGY medium to a cell density at an $OD_{600}$ of 2–6. To induce protein expression, the cells were pelleted at 1,500 g for 5 min at room temperature and were resuspended in 20 ml BMMY medium. Then, the culture was induced every 24 h with 0.5% (v/v) methanol. For large scale expression, cells were grown in 100 ml BMGY to a cell density of $OD_{600}$ at 3–4, then pelleted and resuspended in 1 L BMMY medium. The cultures were induced every 24 h with 0.5% methanol.

After 5 days induction, one liter of medium was harvested by spinning down the cells at 13,000 g for 10 min. The culture supernatant containing ySjCB2 was dialyzed with 3 times volume of PBS and purified by $Ni^{2+}$-NTA affinity-chromatography (GE healthcare, USA) followed by DEAE affinity-chromatography (GE healthcare, USA). The peak fractions were pooled and concentrated using Amicon Ultra-10K device (Millipore Corporation, USA). Purified ySjCB2 was stored in buffer (50 mM NaCl, 20 mM Tris-HCl (pH 8.0), 1 mM EDTA and 5 mM β-mercaptoethanol) at -80°C.

## Deglycosylation of ySjCB2

Deglycosylation of ySjCB2 was performed using endoglycosidase H (New England Biolab Inc., USA) under native and denaturing conditions. Under denaturing condition, twenty μg of ySjCB2 was denatured at 100˚C for 10 min. Then endoglycosidase H was added. The enzymatic reaction was performed at 37˚C for 2 h. For the native condition, the denaturing step was omitted and the enzymatic reaction was performed at 37˚C overnight.

## Determination of ySjCB2 enzyme activity

The enzyme activity assays were performed in triplicate in black, flat-bottomed, 96-well microplate (Thermo Fisher, Shanghai, China) using deglycosylated ySjCB2 (under native condition) in a total volume of 200 μL at 37˚C. Benzyloxycarbonyl-L-phenylalaninyl-L-arginine 4-methyl-coumaryl-7-arginine-MCA (Z-FR-MCA, Sigma-Aldrich, St. Louis, MO, USA) was used as the substrate at a final concentration of 5 μM. Briefly, 50 nM of enzyme solution was added to the assay buffer containing 5 μM Z-FR-MCA and 10 mM dithiothreitol (DTT). The release of fluorescence at the excitation and emission wavelengths of 355 and 460 nm was measured by the Microplate Reader (BioTek, USA). Optimum reaction condition was determined by varying the temperature at 4–65˚C and pH at 3.0–8.0. For the inhibition measurement, different concentrations of E-64 (Sigma-Aldrich, St. Louis, MO, USA) were incubated with ySjCB2 in 100 mM sodium acetate buffer (pH 4.0) and the reaction was initiated by the addition of Z-FR-MCA. For kinetic analysis, ySjCB2 at 50 nM was incubated with varying concentrations of Z-FR-MCA in sodium acetate buffer (pH 4.0) with 10 mM DTT. The release of MCA was monitored over 10 min at 37˚C. The kinetic constants $K_m$ and $V_{max}$ were monitored with GraphPad Prism 6.0 (GraphPad Software, Inc., San Diego, CA).

## Degradation of host proteins by ySjCB2

ySjCB2 (500 nM) was incubated at 37˚C for 18 h with 20–100 μg of collagen, elastin, keratin, fibronectin, immunoglobulin A, immunoglobulin G, immunoglobulin M, complement C3, albumin and hemoglobin (Sigma-Aldrich, St. Louis, MO, USA) respectively, in 100 μL acetic acid-sodium acetate solution (100 mM, pH 4.0). After incubation, 40 μL of the reaction was resolved by SDS-PAGE (4–20% Nupage gel, GenScript, Nanjing, China) and stained with Coomassie Brilliant Blue G250. For the control, protein substrate was incubated without ySjCB2 and analyzed in the same manner.

## Stable isotope dimethyl labeling proteomic analysis of cultivated human epidermis digested with ySjCB2

For the skin digestion experiments, we adapted the cultivated human epidermis from Shanghai ReMed Biotechnology Co., Ltd. Two pieces of cultivated human epidermis was teared from the transwell plate and cut into small pieces, then placed in 1.5 ml microfuge tubes. 200 μL digestion buffer (100 mM sodium acetate, pH 4.0) containing 1.6 μM ySjCB2 or inhibited ySjCB2 was added, the reaction system was mixed by pipetting, and then incubated for 6 h at 37˚C. The inhibited ySjCB2 was produced by incubating 1.6 μM ySjCB2 with 10 μM E-64 for 1 h at room temperature, which was fully inhibited with no activity against Z-FR-MCA. After incubation, the reaction was centrifuged for 15 min at 13,000 g, 4˚C. 100 μL supernatant was collected and lyophilized for the mass spectrometry study. The protein concentration was measured by Bradford Protein Assay (Glory, USA).

The lyophilized samples were re-dissolved to 1 g/L with 100 mM sodium phosphate buffer (pH 6.0). Proteins were reduced with 10 mM dithiothreitol (DTT) at 95˚C for 5 min and

alkylated with 100 mM iodoacetamide at room temperature in dark for 45 min. Next, samples were dimethyl labeled at 25°C in dark with 20 mM formaldehyde solution (light, $CH_2O$; heavy, $CD_2O$) and 20 mM sodium cyanoborohydride. After first 2 h labeling, the reaction was repeated once. After dimethyl labeling, the samples were subjected to filter-aided sample preparation (FASP) method and digested at 37°C with Lys-C for 4 h and trypsin for 12 h. The peptides were analyzed with an easy-nLC 1200 coupled with Orbitrap Fusion Lumos mass spectrometry (Thermo Fisher Scientific, USA). Light labeled peptides and heavy labeled peptides are used to annotate the peptides generated from active ySjCB2 group and inhibited ySjCB2 group, respectively.

The raw data were analyzed by Proteome Discoverer (version 1.4, Thermo Fisher Scientific) and the Human SwissProt database (20160213, 20186 entries) was used. Semi-Trypsin/P was selected as the protease allowing two missed cleavages. Carbamidomethyl on cysteine was set as a fixed modification, and oxidation on methionine was set as variable modification. As for light-labeled samples, dimethylation ($C_2H_6$) on lysine was set as fixed modification and dimethylation ($C_2H_6$) on peptide N-termini was set as variable modifications. For heavy-labeled samples, dimethylation ($C_2H_2D_4$) on lysine was set as fixed modification and dimethylation ($C_2H_2D_4$) on peptide N-termini was set as variable modifications. The incorporated percolator in Proteome Discoverer and the mascot expectation value was used to validate the search results and only the hits with FDR $\leq$ 0.01 and MASCOT expected value $\leq$ 0.05 were accepted for discussion.

### Inhibition of the cercariae invasion

Fifteen C57BL/6 mice were randomly divided into 3 groups (5 mice each), the anti-rSjCB2 IgG, the pre-immune IgG and the water treated groups. The buffer of anti-rSjCB2 IgG and pre-immune IgG was PBS. For anti-SjCB2 IgG treated group, the mice were abdomen shaved, 40 cercariae were counted and put onto a drop of water containing 10 mg/ml anti-rSjCB2 IgG on the cover slip. After 10 min incubation, the cover slide was placed onto the abdomen of the mice with the cercariae-containing side down. For the other two groups, the cercariae were incubated with pre-immune IgG (10 mg/ml) or water in the same manner. The worms were collected 28 days post infection for evaluation of the invasion ability.

## Results

### Bioinformatics characterization of SjCB2 sequence

The open reading frame (ORF) of SjCB2 is 1,047 bp in length and encodes a preproenzyme of 348 amino acids with predicted molecular weight of 39.5 kDa and isoelectric point of 6.48. Based on sequence homology, SjCB2 contains three domains: one transmembrane region (residuals 4–21) at the N-terminal, one pro-peptide domain (residuals 36–76) and the pept_C1 domain (residuals 95–343) which belongs to the cysteine peptidase family C1, subfamily C1A (papain family, clan CA) (Fig 1A). The conserved residuals $Cys^{123}$, $His^{292}$ and $Asn^{312}$ form the catalytic triad of the family C1 of clan CA cysteine peptidases [28]. Also, the sequence contains a unique motif (YWLIANSWXXDWGE) which serves as a hemoglobinase that assists the worm to degrade blood components [28].

Based on multiple sequence alignments, SjCB2 orthologs are found in: (1) other schistosome species-*S. mansoni* and *S. haematobium* sharing 89% and 86% sequence identity, respectively; (2) other trematodes, including *Trichobilharzia regent* (81% identity), *Trichobiharzia szidati* (82% identity), *Clonorchis sinensis* (62% identity) and *Opisthorchis viverini* (65% identity); (3)other cestodes, including *Eudiplozoon nipponicum* (65% identity), *Hymenolepis microstoma* (61% identity) and *Echinococcus multiocularis* (63% identity); (4) one turbellaria,

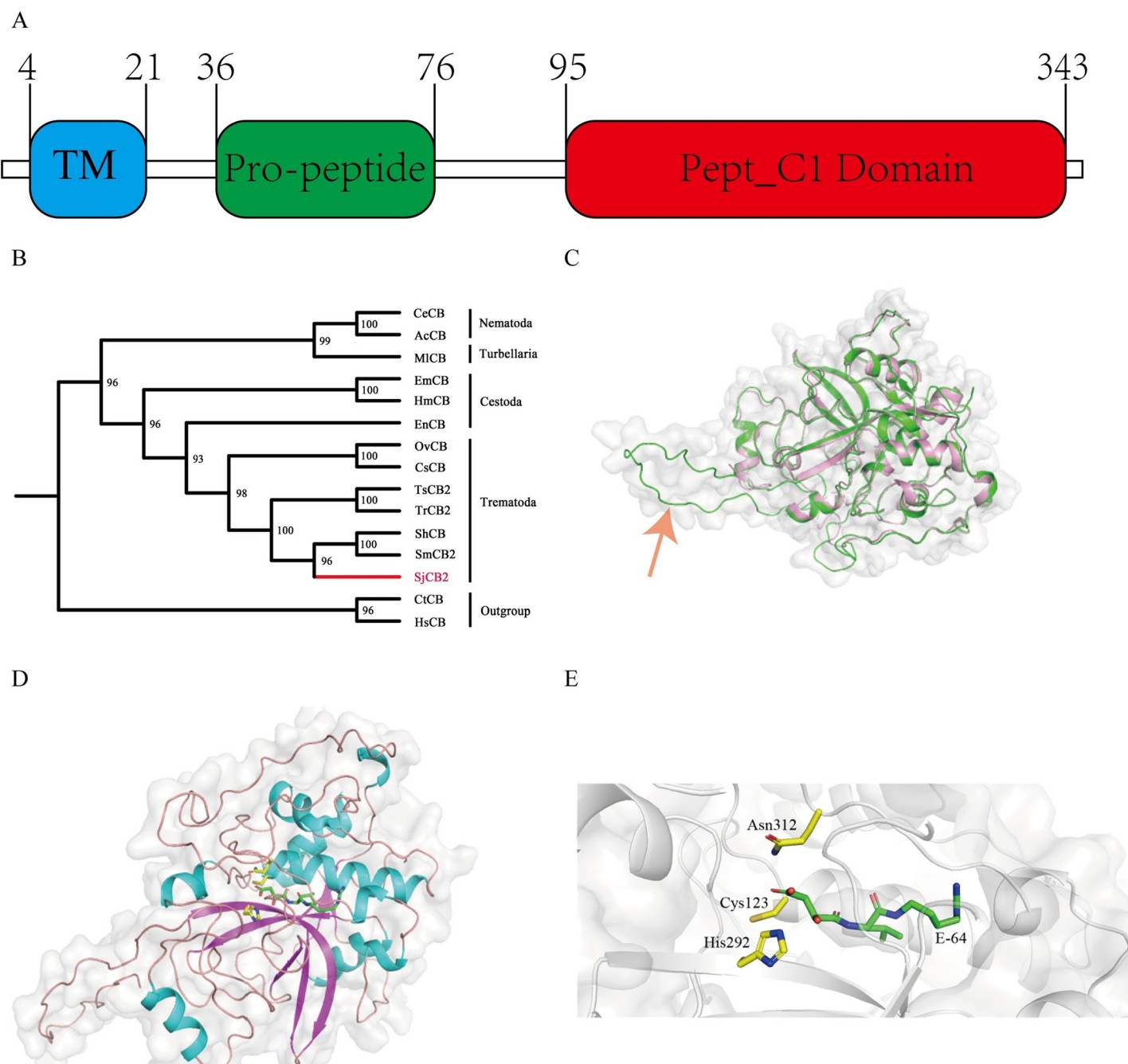

**Fig 1. Bioinformatic analysis and homology modelling of SjCB2.** (A) This diagram shows the domain organization. The N-terminal transmembrane domain, pro-peptide domain and the cysteine pept_C1 domain are depicted in blue, green and red, respectively. Amino acid residue numbers are indicated. (B) The phylogenetic tree constructed by Maximum Likelihood method showing the phylogeny of SjCB2. Cathepsins B of *Capitella teleta* and *Homo sapiens* are set as the outgroup. The following sequences were retrieved from the GenBank: *S. japonicum* cathepsin B2 (SjCB2, ACCESSION: AAO59414.2), *S. mansoni* cathepsin B2 (SmCB2, ACCESSION: XP_018651608.1), *S. haematobium* cathepsin B (ShCB, ACCESSION: XP_012799340.1), *T. regenti* cathepsin B2 (TrCB2, ACCESSION: ABS57370.1), *T. szidati* cathepsin B2 (TsCB2, ACCESSION: ACG50797.1), *C. sinensis* cathepsin B (CsCB, ACCESSION: ABM47070.1), *O. viverrini* cathepsin B (OvCB, ACCESSION: OON22637.1), *E. nipponicum* cathepsin B (EnCB, ACCESSION: AVM38373.1), *E. multilocularis* cathepsin B (EmCB, ACCESSION: BAJ83490.1), *H. microstoma* cathepsin B (HmCB, ACCESSION: CDS27962.1), *M. lignano* cathepsin B (MlCB, ACCESSION: PAA58179.1), *C. elegans* cathepsin B (CeCB, ACCESSION: NP_504682.1), *A. cantonensis* cathepsin B (AcCB, ACCESSION: ADQ57304.1), *C. teleta* cathepsin B (CtCB, ACCESSION: ELT94358.1), *H. sapiens* cathepsin B (HsCB, ACCESSION: Np_001899.1). (C) A superposition of the SjCB2 model (green) and the *S. mansoni* Cathepsin B1 (pink with the PDB code 4I04) in a cylinder representation. The pro-peptide region is indicated by the red arrow. (D) A view from the top on the SjCB2 with the inhibitor E-64. Yellow: Catalytic residues of SjCB2. Green: carbon atoms of E-64. (E) A representation shows the interaction between the SjCB2 active sites and the inhibitor E-64.

*Macrotomum lignano* (59% identity); and (5) two nematodes, *Angiostrongylus carvntonesis* and *Caenorhabditis elegans* sharing 47% and 46% sequence identity, respectively (S1 Fig). Through phylogenetic tree analysis, SjCB2 is closely related to *Schistosoma mansoni* cathepsin B2 (SmCB2) and *Schistosoma haematobium* Cathepsin B (ShCB), sharing the same branch (Fig 1B).

A spatial model of SjCB2 was constructed by homology modeling to study the structure activity and inhibition relationship. As shown in Fig 1C, SjCB2 has the conserved architecture of cathepsin B and cathepsin B-like peptidases composed of a pro-peptide domain and the protease domain. The catalytic amino acid residues, $Cys^{123}$, $His^{292}$ and $Asn^{312}$ are located at the junction between β-barrel domains, a docking model shows the interaction between catalytic residues of SjCB2 and the inhibitor E-64 (Fig 1D and 1E).

## Prokaryotic expression of rSjCB2 and antibody purification

rSjCB2 was expressed in *E. coli* as an insoluble protein with an apparent mass of 65 kDa, which is consistent with the estimated molecular mass of SjCB2 with a GST tag (S2 Fig). A purity over of 90% was obtained after $Ni^{2+}$-NTA affinity-chromatography.

The IgG titers against rSjCB2 was detected by ELISA. The ELISA results clearly showed that the IgG titers of both rabbits had exceeded 1:80,000 (S1 Table). After the GST binding, the IgG titers against GST of the two rabbits were almost the same as the negative control (S2 Table). Therefore, we believed that the GST IgG had been cleared.

## Anti-rSjCB2 IgG identified the native protein in the cercariae extracts

Immunoblotting was used to determine whether anti-rSjCB2 IgG was able to react with the native antigen in the cercariae extracts. Two bands were observed after antibody binding, one was around 50 kDa; the other was around 30 kDa (Fig 2B). No positive band was observed when the *S. japonicum* cercariae extracts sample incubated with the pre-immune rabbit IgG (Fig 2A).

## SjCB2 is localized in the acetabular glands and their ducts of *S. japonicum* cercariae

Immunofluorescence microscopy on fixed cercariae using affinity purified anti-rSjCB2 IgG demonstrated that SjCB2 was located to the acetabular glands and their ducts, which comprised a large percentage of the cercarial head (Fig 3, panel I). In the meantime, DAPI staining and the SjCB2 location seemed to be mutually exclusive (Fig 3, panels I and J). The pre-immune IgG and no primary controls showed no signal (Fig 3, panels A and E, respectively).

## Eukaryotic expression of ySjCB2 and enzyme activity characterization

SDS-PAGE analysis of the protein after purification by $Ni^{2+}$-NTA column revealed a thick smear band ranging between 40 and 90 kDa (Fig 4, lane 1). Only a ~30 kDa band remained after endoglycosidase H treatment (Fig 4, lane 2). To confirm the proteolytic activity of ySjCB2, the conventional enzyme assay was carried out using the fluorogenic peptide substrate Z-FR-MCA. The result showed that the reaction rate increased as the ySjCB2 concentration within a ranged from 0.625 to 320 nM (Fig 5A). The optimal temperature and pH for the maximum activity of ySjCB2 was 37°C at pH 4.0 (Fig 5B and 5C). Furthermore, the enzyme was relatively stable at acidic conditions, but was unstable in neutral conditions. Fig 5D showed the activity of ySjCB2 was inhibited by the cysteine specific inhibitor E-64 in a dose-dependent manner. The kinetic data were obtained through the Michaelis–Menten curve, in which the Y-intercept was the Velocity while the X-intercept was substrate concentration (Fig 5E). According to the curve (Fig 5E), ySjCB2 showed a $Km = 66.29$ μM.

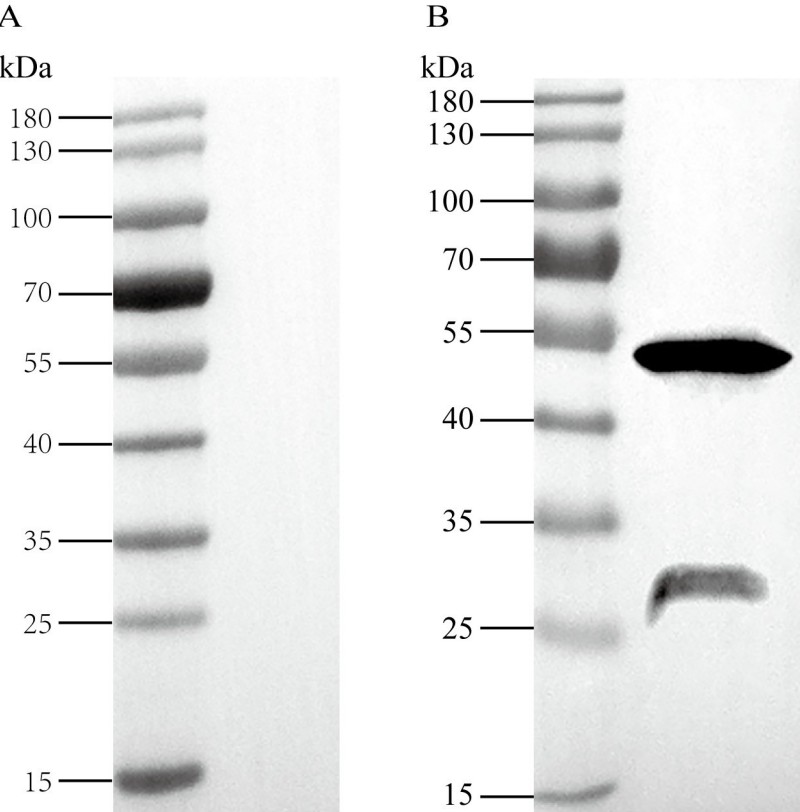

**Fig 2. Identification of native SjCB2.** Protein extracts of *S. japonicum* cercariae were resolved by SDS-PAGE, blotted onto PVDF membrane, and detected by pre-immune rabbit IgG (A) or anti-rSjCB2 IgG (B).

### Processing of natural macromolecular substrates by ySjCB2

To investigate the putative biological roles of SjCB2, the proteolytic activity of ySjCB2 against several host proteins including skin components (collagen, elastin, keratin and fibronectin), immune system components (immunoglobulin A, immunoglobulin G, immunoglobulin M and complement C3), blood components (albumin and hemoglobin) were determined. Collagen, elastin, keratin, fibronectin, immunoglobin A (heavy chain), immunoglobin M, complement C3 and hemoglobin were efficiently degraded by ySjCB2. Immunoglobin G and albumin could also be cleaved, but the effect was not obvious (Fig 6).

### ySjCB2 degraded various proteins of the cultivated human epidermis

Compared to control sample treated with the inhibited ySjCB2, a number of skin proteins were digested by the active ySjCB2. The N terminal of the peptides generated by the active ySjCB2 digestion were labelled by $CH_2O$, and these cleaved proteins were identified as the substrates of ySjCB2 (Table 1). It was found that two cornified layer proteins (involucrin and keratin), five cytoskeleton proteins (actin, keratin, tubulin beta-6 chain, filamin A and plastin), four extracellular proteins (cystatin-B, annexin A1, annexin A2 and heat shock 70 kDa protein 6) were identified as substrates of ySjCB2. Besides, several cell membrane proteins, e. g. alpha-enolase, gamma-enolase and small integral membrane protein 23, as well as two lysosomal proteins, di-N-acetylchitobiase and ganglioside GM2 activator were cleaved by ySjCB2.

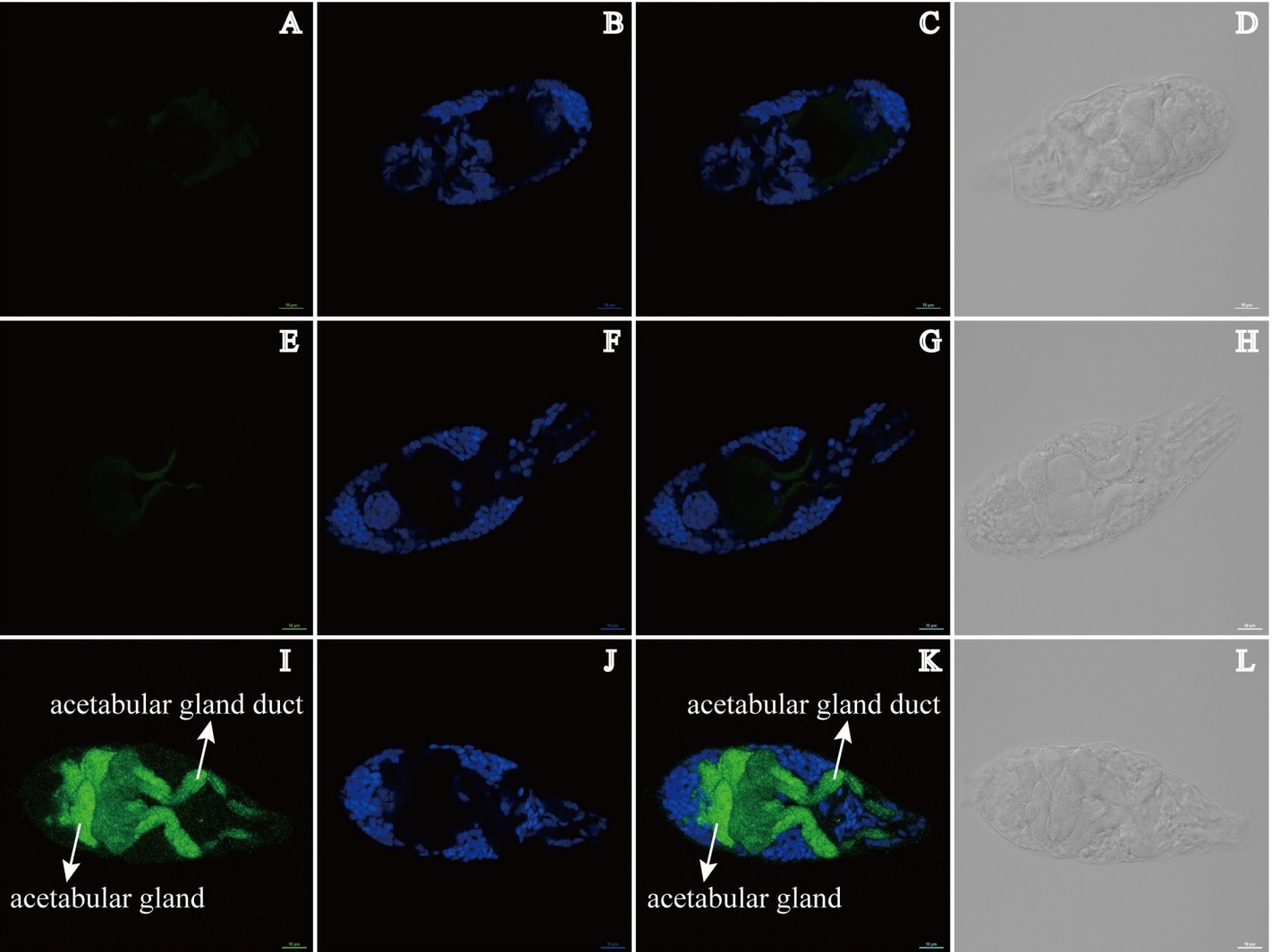

**Fig 3. SjCB2 was located to the acetabular glands and their ducts (pointed out by the white arrows) in the cercariae.** Rabbit polyclonal anti-rSjCB2 IgG and a Goat anti-rabbit Alexa Flour 488 were used to detect SjCB2 in cercariae. (A-D) No primary negative control. The head of the cercariae is facing to the left. (E-H) Pre-immune IgG negative control. The head of the cercariae is facing to the right. (I-L) Anti-rSjCB2 IgG. In panel I, SjCB2 is localized to the acetabular glands and their ducts. Panels B, F, and J are stained with DAPI. Panels C, G and K are merged DAPI and Alexa Flour 488 images. Panels D, H, and L are Differential Interference Contrast (DIC) images for each treatment. Scale bar, 10 μm.

We identified 40 cleavage sites for ySjCB2, the P1 position was enriched for the basic Arg residue, whereas in the P2 position, which was known as the major recognition site in cathepsins [29], mainly identified small aliphatic amino acid Ala (Fig 7A). Fig 7B showed ySjCB2 cleaved tubulin beta-6 chain at VPR/AAL, with an Arg in the P1 position, consistent with their known specificities.

## Antibody of rSjCB2 inhibited the invasion of *S. japonicum* cercariae

Considering that SjCB2 may help cercariae penetrating the mammalian host, an antibody inhibition experiment was performed and the reduction percentage in worm burden was calculated. Compared with the pre-immune IgG control group, a significant worm reduction

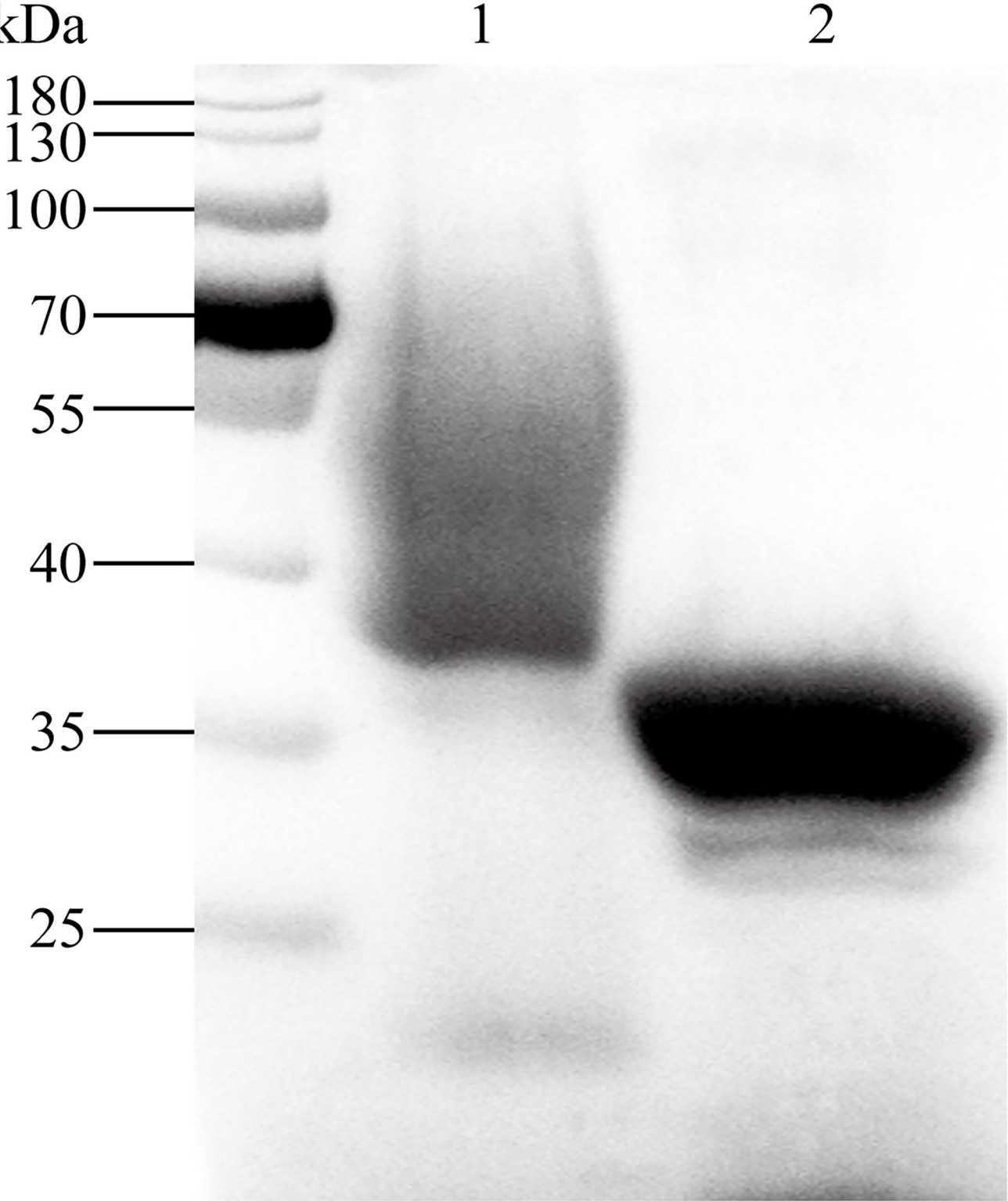

**Fig 4. The Coomassie blue stained 10% SDS–PAGE of ySjCB2 from the yeast culture media purified by Ni²⁺-NTA affinity-chromatography.** The purified ySjCB2 appears as a thick smeared band at 40–90 kDa (lane1). Deglycosylation of ySjCB2 with endoglycosidase H under denaturing conditions resulted in a single band at 30–34 kDa (lane 2). The molecular weight markers are shown on the left side.

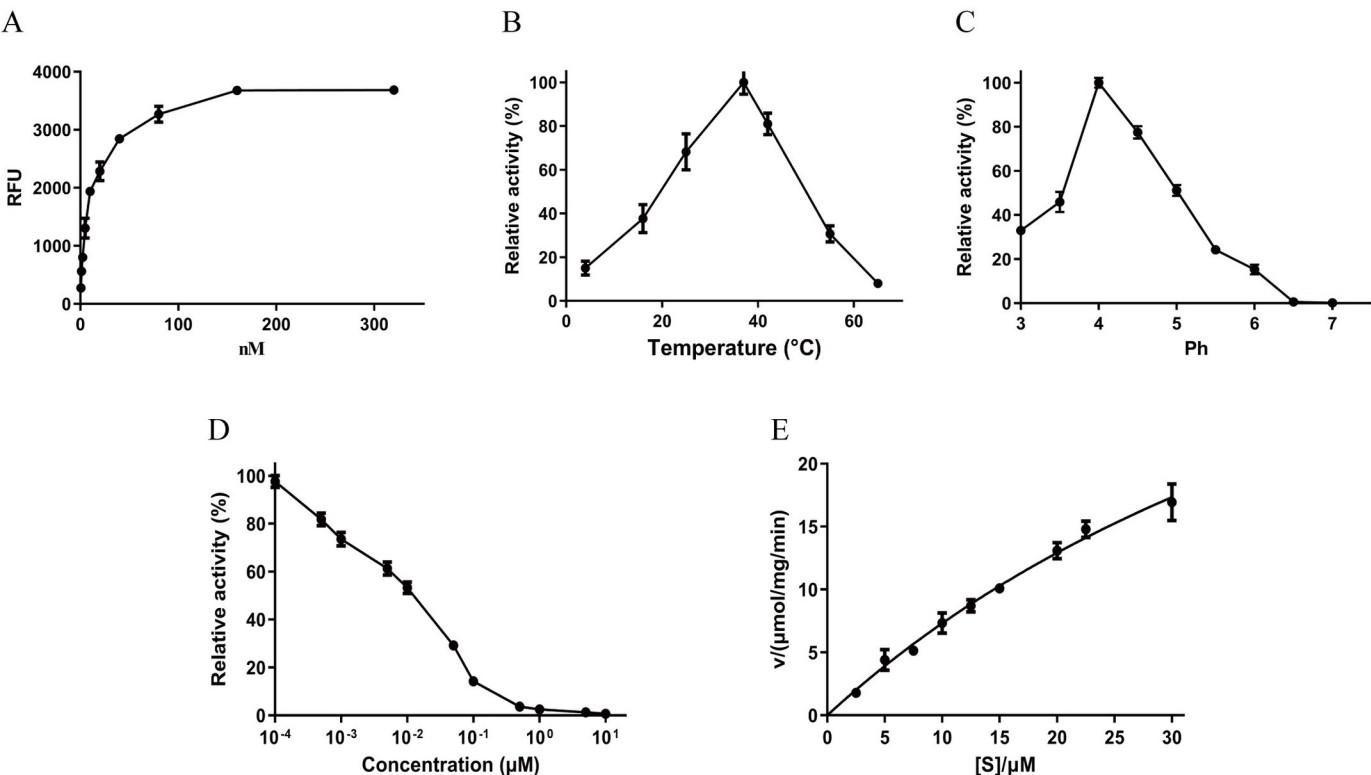

**Fig 5. Biochemical properties of ySjCB2.** (A) Enzyme activity of ySjCB2 was determined by fluorescent substrate Z-FR-MCA under different enzyme concentrations (0.625, 1.25, 2.5, 5.0, 10.0, 20.0, 40.0, 80.0, 160.0 and 320.0 nM). (B) Optimal temperature at pH 4.0. Maximal activity was shown as 100%. (C) Optimal pH, enzyme activity was assayed in various pH buffers ranging from pH 3.0–8.0. Maximal activity was shown as 100%. (D) Inhibition profile for E-64 was determined by incubating ySjCB2 (50 nM) with different concentrations of E-64 in 100 mM acetic acid-sodium acetate (pH 4.0) buffer at room temperature for 30 min. Residual activities (%) were determined using Z-FR-MCA as a substrate. (E) The Michaelis–Menten curve, at 37˚C, pH 4.0.

(22.94%, $p < 0.05$) was observed in the anti-rSjCB2 IgG treated group (Fig 8). There was no significant difference between the water control group and the pre-immune IgG control group (Fig 8).

## Discussion

Skin penetration is the first step for successfully infection in the definitive mammalian host by cercariae. Proteases released from the acetabular glands of the invasive larvae lead to degradation of the epidermis and dermis, evasion of the immune attack and thus successful transmission of the parasite. Therefore, clarifying the function of the proteases involved in the cercariae invasion is essential for understanding the biology of *Schistosoma* and will provide new clues for developing anti-parasitic drugs and vaccines. Previous studies showed that *S. mansoni* cercariae primarily utilize elastases for host invasion [17–19, 30, 31]. Additionally, cathepsin B2 also played a role [32]. Compared with *S. haematobium* and *S. mansoni*, *S. japonicum* is considered evolutionarily more "ancient", which may primarily use cysteine protease in the invasion process [22]. *S. japonicum* contains only one elastase (SjCE2b) and one cathepsin B2 (SjCB2), which both are highly homologous to *S. mansoni* [32, 33]. Since no active SjCB2 or SjCE2b was obtained before, their roles in the *S. japonicum* cercariae invasion process remain unknown. In this report, we have successfully expressed the active SjCB2 with *P. Pastoris* and confirmed its role in the parasite penetration process.

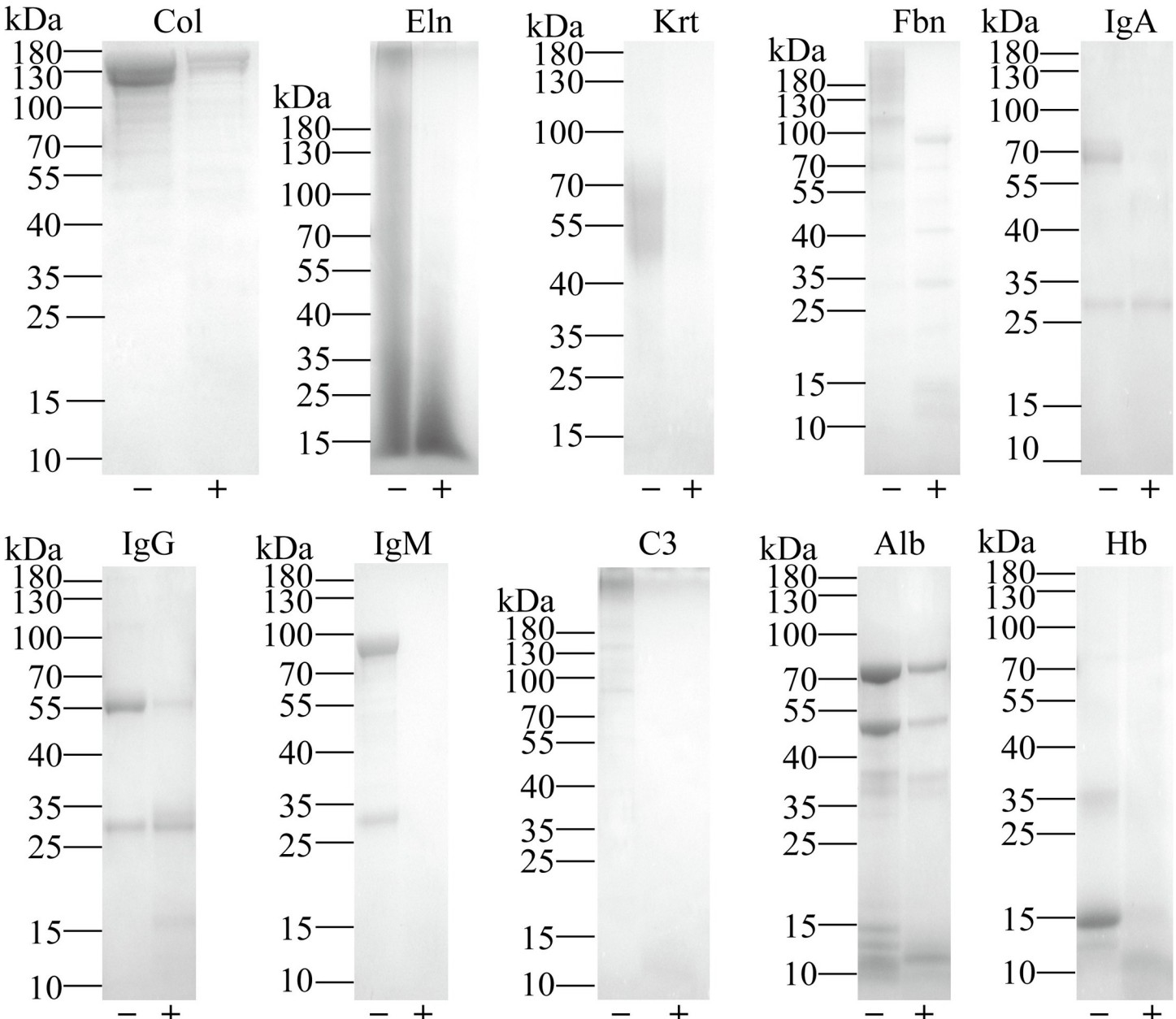

**Fig 6. Digestion of selected natural substrates.** Proteolytic activity of ySjCB2 against main host skin components, collagen (Col), elastin (Eln), keratin (Krt) and fibronectin (Fbn); immune system components, immunoglobulin A (IgA), immunoglobulin G (IgG), immunoglobulin M (IgM) and complement C3 (C3) and blood components, albumin (Alb) and hemoglobin (Hb). Substrates were incubated in 100 mM acetic acid-sodium acetate (pH 4.0) buffer with 10 mM DTT in the presence (+) or absence (-) of ySjCB2 for 18 h at 37˚C and analyzed by 10% SDS-PAGE. Molecular weight marker of each substrate was showed on the left side.

In whole body extracts of *S. japonicum* cercariae, the anti-rSjCB2 IgG recognized two bands at ~50 kDa and ~30.5 kDa. The 50 kD band likely corresponds to the zymogen, and its higher molecular weight might be the result of post-translational modification, most likely the glycosylation. The 30.5 kD protein likely represents the native mature SjCB2. This observation is similar to the cathepsin B2 and B3 from *Fasciola gigantica*, which also showed two forms of the enzyme in the newly excysted juveniles body extracts [34–36].

**Table 1. Substrates of SjCB2 detected in the cultivated human epidermis.**

| Accession number | Protein name |
| --- | --- |
| Cornified layer protein | |
| Q04695 | Keratin, type I cytoskeletal |
| P07476 | Involucrin |
| Cytoskeletal protein | |
| P60709 | Actin |
| Q04695 | Keratin, type I cytoskeletal |
| Q9BUF5 | Tubulin beta-6 chain |
| P21333 | Filamin |
| P13797 | Plastin |
| Extracellular protein | |
| P04080 | Cystatin-B |
| P04083 | Annexin A1 |
| P07355 | Annexin A2 |
| P17066 | Heat shock 70kD protein 6 |
| Cell membrane protein | |
| P06733 | Alpha-enalose |
| P09104 | Gamma-enalose |
| A6NLE4 | Small integral membrane protein 23 |
| P22735 | Protein-glutamine gamma-glutamyltransferase K |
| Q6UWN5 | Ly6/PLAUR domain-containing protein 5 |
| P46940 | Ras GTPase-activating-like protein |
| Lysosomal protein | |
| Q01459 | Di-N-acetylchitobiase |
| P17900 | Ganglioside GM2 activator |
| Cytoplasm/ Nucleus protein | |
| Q01844 | RNA-binding protein |
| Q13404 | Ubiquitin-conjugating enzyme E2 variant 1 |
| Q9UL68 | Myelin transcription factor 1-like protein |
| P26599 | Polypyrimiding tract-binding protein 1 |
| P48637 | Glutathione synthetase |
| Q07955 | Serine/arginine-rich splicing factor 1 |
| P14618 | Pyruvate kinase PKM |
| P04792 | Heat shock protein beta-1 |
| P14678 | Small nuclear ribonucleoprotein-associated proteins B and B' |
| P09211 | Glutathione S-transferase P |
| Q01469 | Fatty-acid binding protein 5 |

Using immunofluorescence, the native SjCB2 was detected in the acetabular glands and along the gland ducts. There are two pairs of pre-acetabular glands and three pairs of post-acetabular glands in the cercariae head. These unicellular glands have large fundi that fill the body cavity and ducts anteriorly through the muscle cone, terminating at the oral sucker [37]. During the penetration process, *Schistosoma* cercariae will secrete the contents of this two sets acetabular glands. The released mucus-like substances and proteolytic enzymes will help cercariae attaching host's skin, disrupting the skin structure, removing the cercarial glycocalyx during the stage transition and protecting against the host immune attack [38]. Therefore, our finding offers a possibility of this enzyme in aiding cercariae penetrating their definitive mammalian host.

A

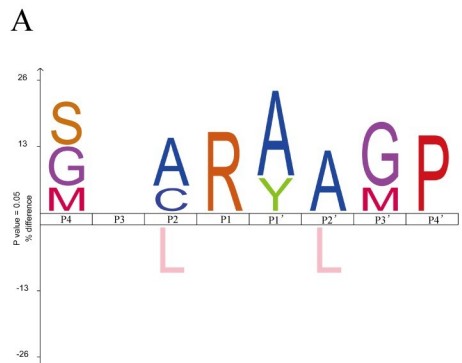

B

Tubulin beta-6 chain

```
1-58      MREIVHIQAGQCGNQIGTKFWEVISDEHGIDPAGGYVGDSALQLERINVYYNESSSQK
59-115    YVPR/AALVDLEPGTMDSVRSGPFGQLFRPDNFIFGQTGAGNNWAKGHYTEGAELVDA
116-173   VLDVVRKECEHCDCLQGFQLTHSLGGGTGSGMGTLLISKIREEFPDRIMNTFSVMPSP
174-233   KVSDTVVEPYNATLSVHQLVENTDETYCIDNEALYDICFRTLKLTTPTYGDLNHLVSA
234-290   TMSGVTTSLRFPGQLNADLRKLAVNMVPFPRLHFFMPGFAPLTSRGSQQYRALTVPEL
291-348   TQQMFDARNMMAACDPRHGRYLTVATVFRGPMSMKEVDEQMLAIQSKNSSYFVEWIPN
348-406   NVKVAVCDIPPRGLKMASTFIGNSTAIQELFKRISEQFSAMFRRKAFLHWFTGEGMDE
348-447   MEFTEAESNMNDLVSEYQQYQDATANDGEEAFEDEEEEIDG
```

**Fig 7. Cleavage specificities of SjCB2.** (A) The cleavage sites for SjCB2 are represented as iceLogos. (B) Protein sequence of tubulin beta-6 chain, the cleavage site is pointed by inserted triangle.

Due to the failure to obtain the active SjCB2 in *E. coli*. We adopted the yeast expression system, *P. Pastoris*. This system is well-known for the eukaryotic protein expression, the post-translational modifications and the capacity for protein secretion [39, 40]. Besides, it has been used to the expression of multiple isoforms of cathepsin B from *Schistosoma* [23, 25] and *Fasciola* [35, 36, 41]. The predicted sizes of the proSjCB2 and the mature SjCB2 are 36.8 kD and 30.5 kD, respectively. The recombinant SjCB2 secreted by *P. pastoris* migrated as a smeared band ranging from 40–90 kDa (Fig 4, lane 1). Since *Pichia* is known to glycosylate secreted proteins and the proenzyme has three potential N-linked glycosylation sites at positions 47, 186 and 218. It was reasonable that this higher molecular smeared band was most likely due to abnormal high levels glycosylation by yeast cells. Treated of the denatured recombinant ySjCB2 with endoglycosidase H resulted in a quantitative conversion of the recombinant protein to a 30–34 kDa species which most likely represents the mature SjCB2 protein (Fig 4, lane 2). This suggests that the recombinant proSjCB2 is capable of auto-catalytic activation and maturation in the *Pichia* induction medium.

Z-FR-MCA, a widely used and cysteine protease-specific substrate [28, 42, 43], was used to test the biochemical characteristics. The optimal temperature for ySjCB2 was 37°C, which is the physical temperature of human body. ySjCB2 can cleave the substrate under pH 3.0–6.5, while maximum activity was assayed at pH 4.0, suggesting that ySjCB2 was active under acidic conditions. In most cases the pH range for normal human skin is said to be between 5.4–5.9 [44], so we predict that SjCB2 will be active after its release into the human skin. Besides, cysteine proteases also involve in blood digestion to provide nutrition for the parasites [45]. The pH of the *Schistosoma* gut has been measured at 5.0 or lower [46], which is also very suitable for SjCB2. Previous studies showed that SjCB2 was also highly expressed in the adult worms [47, 48], and we found that ySjCB2 can degrade host hemoglobin and albumin. These data suggest a role for SjCB2 in degrading blood proteins to absorbable nutrients to support the female worms' egg production. The peptidolytic activity of ySjCB2 was efficiently inhibited by the general clan CA cysteine peptidase inhibitor E-64 [49] at micromolar concentration. With the substrate we used, a $K$m value of 65.29 μM was determined, which was close to SmCB2 [25]. The activity and inhibition data of ySjCB2 presented here corroborate previously reported results on cathepsins B of *Schistosoma* and *Trichobilharzia* flukes [25, 28, 50].

To establish the infection, *Schistosoma* cercariae need to pass through the epidermis and the dermis of the skin, then get into the blood or lymphatic vessel [51]. Epidermis, the outermost layer of human skin [52], is the first physical barrier for cercariae. The epidermis is an avascular stratified squamous epithelium mainly containing keratinocytes, but also

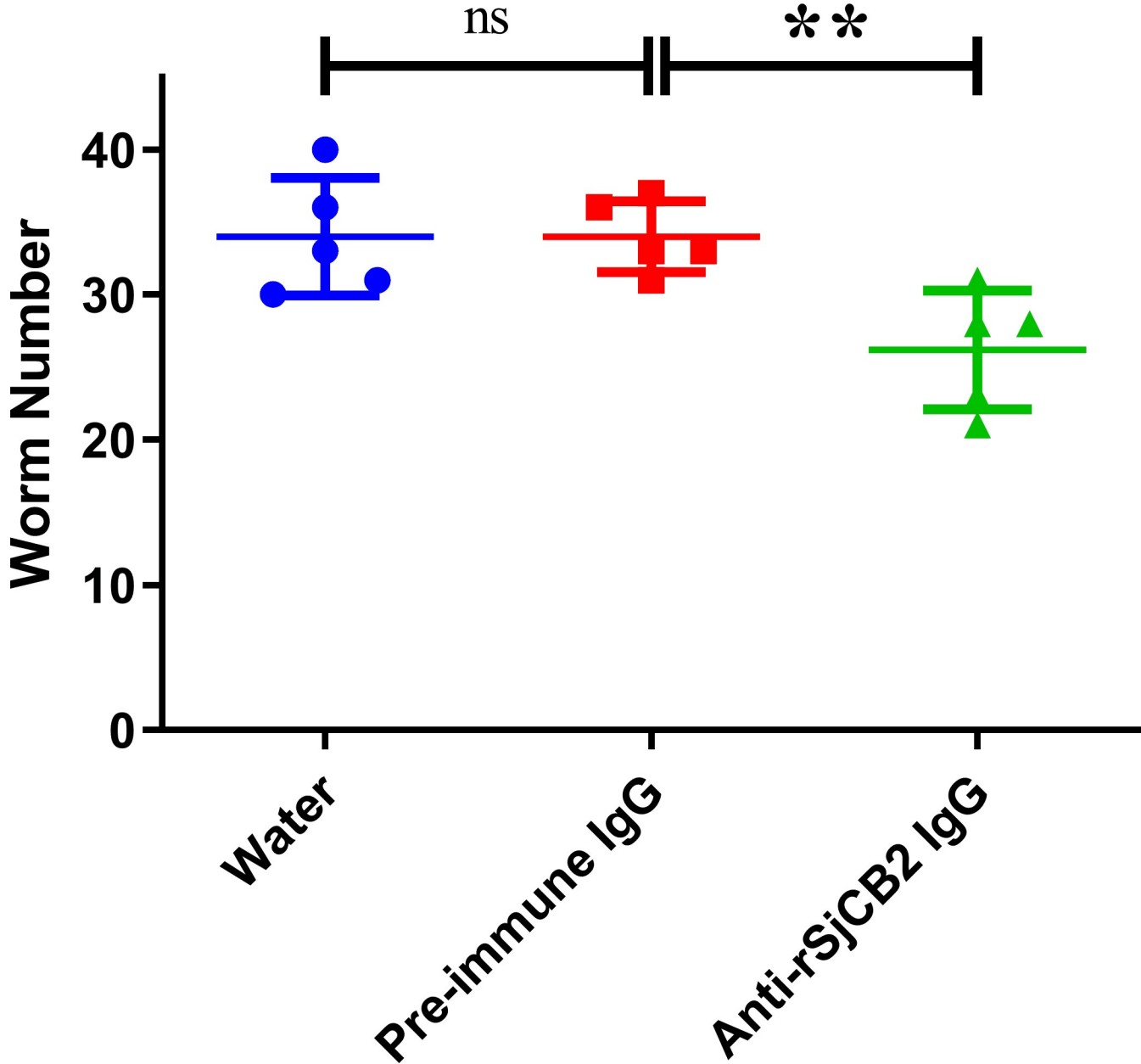

**Fig 8. Inhibition of cercariae invasion.** Mice were experimentally challenged with 40 *S. japonicum* cercariae and adult worms were collected 28 days post infection. Statistically significant difference (** = p-value<0.01, * = p-value<0.05).

melanocytes, Langerhan cells and Merkel cells [53]. Most of the defensive functions of the epidermis localize to the stratum corneum, also called the cornified layer. Two cornified envelope proteins of the corneocytes, keratin and involucrin, can be degraded by ySjCB2, indicating that SjCB2 is able to digest the cornified layer. The cytoskeleton consists of three main types of fibrils, which are actin filaments, intermediate filaments and microtubules [54]. Actin, keratin and tubulin are subunit of these three fibrils, respectively. Filamin promotes orthogonal branching of actin filaments and binds many cellular constituents [55]. Plastin is responsible for the actin-bundling [56]. Addition of ySjCB2 lead to the cleavage of all these five cytoskeleton proteins, demonstrating that this protease can lyse the keratinocytes by disrupting the

cytoskeleton structure. This will cause a great damage to the epidermis. In addition, cystatin-B, a reversible inhibitor of cathepsin B [57], was degraded, showing that SjCB2 can resist the inhibition by host cystatin. Resistance of host cystatin by cathepsin B was also found in *F. hepatica* [58]. These results imply that SjCB2 is involved in the *in vivo* degradation of the epidermis in order to facilitate cercariae penetration.

Next, cercariae will encounter the dermis, the structural foundation of the skin. The dermis is located below the epidermis and consists of collagen and elastin fibers [59]. *In vitro* degradation assays showed that ySjCB2 can digest collagen and elastin efficiently, indicating that this protease also can disrupt the structure of the dermis.

Cathepsin B of human was the first discovered that can cleave elastin [60], but it was found that both TrCB2 and SmCB2 also can cleave elastin later [28, 32]. Besides, ySjCB2 can degrade the fibronectin, a major component of blood clotting. Adult worms pair and reside in blood vessels, which will disrupt blood flow, activate the platelet and cause blood coagulation [61]. However, *Schistosoma* adopt a variety of methods to inhibit blood clot formation and/or promote blood clot lysis [62]. Recently, we discovered that SjCB2 is presented in adult worms' ES products (S3 Fig), so we speculate it may be involved in thrombus degradation. Furthermore, many components of immune system (IgA, IgG and IgM) and one component of complement system (complement C3) were digested by ySjCB2. This suggests that this protease may help parasite evading the host immune attack by proteolysis of attaching immunoglobulins and complement factors. There are number of reports revealed that cathepsins B from flatworms are able to degrade immune-related proteins. SmCB2 of *S. mansoni* can degrade IgG and complement C3 [32]; FhCB2 of *F. hepatica* is able to digest IgG [63]; EmCBP1 and EmCBP2 of *E. multilocularis* and four cathepsins B (CsCB1,CsCB2,CsCB3,CsCB4) of *C. sinensis* are capable of cleaving IgG [43]. Hence cathepsin B is a general protease utilized by parasites for immune invasion.

Proteolytic enzymes released by cercariae are the key factors for skin penetration. Therefore, blocking or inhibiting important protease is an effective way to intervene cercarial invasion. Previous works showed that use of specific inhibitors targeting serine protease could block cercarial invasion of human skin [64, 65]. Chemical inhibitors usually target family rather than a single protease, so we can't evaluate the contribution of a specific protease. Since SjCB2 was released by the *S. japonicum* cercariae [22], we used the anti-rSjCB2 IgG to bind the native protein in the cercarial secretion and observed the worm number had a 22.94% decline, directly proved that SjCB2 was involved in the skin penetration. However, less than a quarter of the contribution indicates that other proteases are involved in this process, and further studies will be needed.

In conclusion, we expressed, and biochemical and functionally characterized a cysteine protease, SjCB2. This protease plays pivotal roles in *S. japonicum* cercariae invasion, a process that is essential for the parasite infects the definitive mammalian host. It's considered to be a multi-functional enzyme, capable of digesting blood components, degrading thrombus and evading the immune attack, contributing to parasite survival. Future research can be concentrated on digging out other important proteases involved in the *S. japonicum* cercarial invasion and designing a vaccine or drug against SjCB2.

## Supporting information

**S1 Fig. Sequence alignment of SjCB2, the percentage amino acid identity of SjCB2 with each Cathepsin B is shown at the end of the alignment.** Predicted N-glycosylation sites are highlighted in grey. The active site residues are indicated by the five-pointed star, Cys123 and His292, forming a catalytic dyad; Gln312, preceding the catalytic Cys123, involved with the

formation of the oxyanion hole; and an Asn117 residue which orients the imidazolium ring of the catalytic His292. The S2 subsite residues are indicated by the solid circle, they represent the dominant substrate specificity subsite of papain-like cysteine proteases. Cys residues of the pept_C1 domain that are predicted to form the disulfide bond are indicated by the same number.
(TIF)

**S2 Fig. Expression and purification of recombinant SjCB2 in *E. coli*.** Lane 1: soluble fraction from *E. coli* cell lysates; Lane 2–3: column flow through; Lanes 4–8, 50 mM imidazole eluent; Lanes 9–12, 250 mM imidazole eluent; M, PageRuler Prestained Protein Ladder (Thermo Fisher Scientific). Arrowhead indicates rSjCB2.
(TIF)

**S3 Fig. SjCB2 was identified in the 42 dpi worms' ES products.** ES proteins of 42 dpi adult worms were resolved by SDS-PAGE, blotted onto PVDF membrane, and detected by pre-immune rabbit IgG (A) or anti-rSjCB2 IgG (B).
(TIF)

**S1 Table. Determination the titer of rabbit polyclonal antiserum against rSjCB2 by indirect ELISA.**
(XLSX)

**S2 Table. Determination of GST IgG titer in anti-rSjCB2 IgG after treatment by indirect ELISA.**
(XLSX)

## Acknowledgments

We greatly appreciate John Dalton and Dave Dyer for the linguistic modification.

## Author Contributions

**Conceptualization:** Wei Hu.

**Data curation:** Bingkuan Zhu, Fang Luo, Yi Shen, Wenbin Yang, Chengsong Sun, Yongdong Li.

**Formal analysis:** Bingkuan Zhu, Fang Luo, Yi Shen, Wenbin Yang, Chengsong Sun, Yongdong Li, Wei Hu.

**Funding acquisition:** Wei Hu.

**Investigation:** Bingkuan Zhu, Yi Shen.

**Methodology:** Bingkuan Zhu, Yi Shen, Jipeng Wang, Xumin Zhang, Yongdong Li, Wei Hu.

**Project administration:** Yongdong Li, Wei Hu.

**Resources:** Jian Li, Xiaojin Mo, Bin Xu, Yongdong Li.

**Software:** Bingkuan Zhu, Fang Luo.

**Supervision:** Yongdong Li, Wei Hu.

**Validation:** Bingkuan Zhu.

**Visualization:** Bingkuan Zhu.

**Writing – original draft:** Bingkuan Zhu.

**Writing – review & editing:** Wei Hu.

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
