## [Decision Letter · Decision Letter 0]

5 Aug 2020

Dear Professor Hu,

Thank you very much for submitting your manuscript "Schistosoma japonicum cathepsin B2 (SjCB2) facilitates parasite invasion through the skin" for consideration at PLOS Neglected Tropical Diseases. As with all papers reviewed by the journal, your manuscript was reviewed by members of the editorial board and by several independent reviewers. The reviewers appreciated the attention to an important topic. Based on the reviews, we are likely to accept this manuscript for publication, providing that you modify the manuscript according to the review recommendations. 

Sincerely,

Katja Fischer

Guest Editor

Michael Hsieh

Deputy Editor

Reviewer's Responses to Questions

**Key Review Criteria Required for Acceptance?**

**Methods**

-Are the objectives of the study clearly articulated with a clear testable hypothesis stated?

-Is the study design appropriate to address the stated objectives?

-Is the population clearly described and appropriate for the hypothesis being tested?

-Is the sample size sufficient to ensure adequate power to address the hypothesis being tested?

-Were correct statistical analysis used to support conclusions?

-Are there concerns about ethical or regulatory requirements being met?

Reviewer #1: In the section of Antiserum of rSjCB2 inhibited the invasion of S. japonicum cercariae.

It seems to me that the rSjCB2 antiserum includes antibodies against GST and SjCB2, thus it does not provide a convincing result for the function of SjCB2, as the GST protein is also S. japonicum originated. Also, the design of the experiment is questionable as water but not buffer was used for the binding reaction between antibodies and the parasite-released SjCB2. Antibody-antigen binding usually is most efficient in aqueous buffers at physiological pH and ionic strength, such as in phosphate-buffered saline. Without buffer, the affinity between antibodies and antigen is low. The authors should redesign and re-perform the experiment.

Reviewer #2: Methods are appropriate to study aims. No real concerns, however clarification is required.

Expression and purification of ySjCB2- Presumably deglycosylation is not part of purification process? If so, please write the method for deglycosylation separately from purification.

Degradation of host proteins- It is noted that 500 nM of purified ySjCB2 was used is digestion of tested substrates. This is approximately 5 fold higher than the maximum enzyme concentration used in the enzyme activity assay (80-100 nM, Fig 5A). Did the authors test out a range of enzyme concentrations for digestion of all substrates tested? Why did the authors not include inactivated ySjCB2 as a negative control, eg. By using E-64 or heat inactivation as in the cultivated human digest assay.

Inhibition of the cercariae invasion- How did the authors decide to use 10 min incubation period with SjCB2 antiserum? Did the authors test a range of incubation times? Was the serum incubated with cercariae at RT or at 37�C?

Reviewer #3: (No Response)

**Results**

-Does the analysis presented match the analysis plan?

-Are the results clearly and completely presented?

-Are the figures (Tables, Images) of sufficient quality for clarity?

Reviewer #1: (No Response)

Reviewer #2: General comment – Authors should use full name or include explanation of acronyms of enzymes when referring to them for easier reading. For example, Line 389- SjCB2 is closely related to SmCB2 and ShCB sharing the same branch (Fig 1B). The explanation for what SmCB2 and ShCB are only described in the figure legend. e.g. in line 407. The use of genus and species should be consistent, with the name written in full when used for the first time. Please amend throughout the manuscript as appropriate.

Fig 3 – Please include arrows indicating the cercariae heads panel D, H, L. It would be visually helpful if there is a diagram of acetabular glands and their ducts of S. japonicum cercariae or indication of these structures in the scans.

Is purified recombinant ySjCB2 used in this study in glycosylated form as produced in Pichia pastoris? Will this affect its enzymatic activity?

Was the sequence of purified ySjCB2 confirmed by N-terminal sequencing?

Degradation of host proteins- Fig 6. To standardize the results, all samples should be run on the same gel and includes protein ladder to mark sample size. Fbn, IgAand C3 gels are very faint to visualize.

Reviewer #3: (No Response)

**Conclusions**

-Are the conclusions supported by the data presented?

-Are the limitations of analysis clearly described?

-Do the authors discuss how these data can be helpful to advance our understanding of the topic under study?

-Is public health relevance addressed?

Reviewer #1: Line 135-6, whether it is appropriate to state ‘and also offered a new target for novel anti-schistosomal therapeutics’ is questionable, as no functional studies on SjCB2 in the intra-mammalian developmental stage have been carried out in the current work.

Reviewer #2: Line 553-555 – Please confirm the author’s statement accurately reflects what is reported in the referenced article 32.

ySjCB2 appears to be promiscuous with a wide range of substrates, ranging from collagen, elastin, fibrinogen, IgM, C3, haemoglobin, etc. Are there any other cercariae’s cathepsin Bs that digest a wide range of proteins?

Could the authors include possible reasons for many of the various skin components individually tested with ySjCB2 did not appear in the results from cultivated human epidermis?

Only Keratin was apparent in the analysis in Table 1, but not collagen, fibronectin etc.

Could the authors speculate that if cercariae were treated with general cysteine protease inhibitor such as E-64, will reduction in cercariae invasion in the mouse experiment be more profound that using anti-ySjCB2 serum? 

Line 670- 677. Considering that there would be a family of proteases required to work together for host invasion, evident by only 22% reduction in cercariae infection with anti-ySjCB2 treatment in mice, would targeting specific protease such as ySjCB2 be useful in developing anti-invasion therapeutics?

Reviewer #3: (No Response)

**Editorial and Data Presentation Modifications?**

Reviewer #1: Minor Revision

Reviewer #2: (No Response)

Reviewer #3: (No Response)

**Summary and General Comments**

Reviewer #1: In this study, Zhu et al. reported bioinformatics characterization, localization, biochemical features and functions of a cathepsin B cysteine protease derived from Schistosoma japonicum, SjCB2, and explored the function of the enzyme by cercariae invasion experiment in a murine model. The current study is a piece of work that follow up the previous studies on SmCB2, however, it does provide some feature of its own based on robust data. The results and discussion were presented in an appropriate manner.

Major concerns

1. It is quite confusing that the authors produce the anti-rSjCB2 antibody using a GST-fused protein, which requires additional step to get rid of anti-GST antibody later on prior to application? So why not express the protein with a His-tag only in E. coli, or use the recombinant SjCB2 protein produced in P. pastoris. Line 200, the author stated that ‘incubating the serum with purified GST protein’, I wonder whether the affinity-purified IgG was further incubated with GST protein, otherwise, in the line 201, it should be described as ‘without anti-GST antibody (not only IgG, but also other isotypes). In the current format, the recombinant SjCB2 is actually GST-rSjCB2, the IgG titers tested in the two immunized rabbits are against GST-rSjCB2.

2. In the methods section, In the immunoblotting, the authors stated that ‘anti-rSjCB2 polyclonal IgG diluted 1:1000’, while in the immunofluorescence assay, SjCB2 antiserum in blocking solution with a concentration of 1 μg/ml. ‘anti-rSjCB2 polyclonal IgG’ or ‘SjCB2 antiserum’ are quite ambiguous without appropriately described in the methods section.

In the results, the authors stated that ‘Immunoblotting was used to determine whether rabbit antiserum’, and ‘Immunofluorescence microscopy on fixed cercariae using affinity purified polyclonal antibodies raised against rSjCB2’, which are conflict with those described in the methods. Noting that the purified polyclonal antibodies and antiserum are different concepts.

Same problem: Line 449, using affinity purified polyclonal antibodies while in Line 456, rabbit polyclonal primary antibody was used. What are you used exactly? For the later one, purified or not purified? 

Line 441, ‘sample incubated with the pre-immune rabbit serum’ while Line 444 ‘visualized by normal rabbit IgG’

Check these phrases throughout the manuscript. Rigorous and clear expression is definitively needed.

3. In the section of Antiserum of rSjCB2 inhibited the invasion of S. japonicum cercariae.

It seems to me that the rSjCB2 antiserum includes antibodies against GST and SjCB2, thus it does not provide a convincing result for the function of SjCB2, as the GST protein is also S. japonicum originated. Also, the design of the experiment is questionable as water but not buffer was used for the binding reaction between antibodies and the parasite-released SjCB2. Antibody-antigen binding usually is most efficient in aqueous buffers at physiological pH and ionic strength, such as in phosphate-buffered saline. Without buffer, the affinity between antibodies and antigen is low. The authors should redesign and re-perform the experiment. 

4. Which form of ySjCB2 was used for the determination of enzyme activity and degradation of host proteins? The deglycosylated one or the one without deglycosylation?

5. I suggest that authors should determine the expression profile of the gene throughout different developmental stages, which can further provide the insights on the molecule. If not at least discuss the work by Liu et al (PLoS Comput Biol. 2014 Oct; 10(10): e1003856.), in which it shows that the relatively highest expression of SjCB2 is in the adult male worms, indicating a different functional cue for the gene.

6. Line 96, Adult schistosome worms reside 

Line 97-98, noting that different Schistosoma species have different egg yields per day per pair. Also, Line 98, ‘particularly the liver,’ refers to the infection of S. mansoni and S. japonicum, but not S. haematobium

7. Lines 115, serine protease, and cercarial elastase. Noting that the references 17-21 also include the work from S .japonicum, while the sentence refers the enzymes from S. mansoni

Minor ones:

Line 123, may be both

Line 128, a full name required for P. pastoris,

Line 130, blood circulatory system? 

Line 132, Immunochemistry or Immunofluorescence?

Line 286, what does it mean that ‘S. japonicum cercariae were isolated from Anhui province, China’

Line 311, anti-GST antibody

Line 369, the open reading frame (ORF) of SjCB2 is 1,048 bp in length and encodes a preproenzyme of 348 amino acids. 348 multiply 3 is 1044bp, if add the stop codon is 1047bp.

Line 378, degrade blood components?

Line 444, detected by normal?

Line 477, data were

Line 498, natural macromolecular substrates?

Line 642, facilitate cercariae penetration?

Line 682, blood components

Reference should be follow the style of the journal, and all genus names in references should be italic.

Reviewer #2: This study investigate the role of cathepsin B2 (SjCB2) of Schistosoma japonicum for host invasion. Recombinant SjCB2 was expressed, purified and utilized to address its function. SjCB2 appeared to digest a wide range of host proteins including skin, blood and innate immune components. However, further studies should be conducted to confirm these findings and to investigate SjCB2 specific function.

Reviewer #3: The study has been designed well and have addressed key aspects of the potential activities of the proposed parasitic protease with the help of previous publications. 

Major comments:

Line 187 - Imidazole concentrations of the Ni elution buffers were not mentioned and no information of whether the eluted proteins are dialysed into a new buffer. Depending on that, if the imidazole concentration exceeds the bradford assay comparability, the use of bradford assay may not be accurate for the protein quantification. Dialysis into a new buffer could also result in protein precipitation. So its better to address this in the manuscript in a sensible manner.

Line 175 - Methods doesn't mention about the inclusion of the GST tag in the cloning. But later (Line) mentioned about the presence of a GST tag. It should be mentioned in the cloning section. 

Line 503 - Author has mentioned "All the selected protein substrates were efficiently degraded by ySjCB2". This statement need more clarification. Its has only been done with visual detection and has not used any band intensity measuring software, therefore the level of efficiency is very wage. In fact the band in the albumin and IgG are has considerably similar intensities between the - and +. In addition, mentioning of the resulted fragments in an denaturing gel for each of the protein used with their respective molecular weight would help to explain the resulted bands in each of + lanes. Therefore this section of results need to be thoroughly explained. Furthermore, addition of a protein ladder for the image is required. 

Line 569 - The explanation given for obtaining two bands for rSjCB2 for the whole body extract would be a one possibility. The other possibility might be, eventhough there is single SjCB2 there could be more Sj cathepsin Bs. Reference 35 suggests there might be 3. Can authors explain how they can exclude the cross reactivity of the rSjCB2 antibody with closely related Sj other cathepsins. This comment also related to the localisation results, but that could be excluded since the acetabular glands are the only localised site. 

Minor comments:

Line 195 - The site of infection is needed to be mentioned. 

Line 198 - Correction from G-separate into G-sepharose

Line 222 - homogenisation method and conditions or a reference to a followed method is needed.

Line 264 - Method for the propergation of the P. pastoris X-33 strain in brief or a reference to a followed method is needed.

Line 266 - Electroporation conditions are needed. 

Line 267 - Please check the font size

Line 291 - SI unit of for the microliter need to be corrected.

PLOS authors have the option to publish the peer review history of their article (what does this mean?). If published, this will include your full peer review and any attached files.

Reviewer #1: No

Reviewer #2: No

Reviewer #3: No
---

## [Editor Report · Decision Letter 1]

22 Sep 2020

Dear Professor Hu,

We are pleased to inform you that your manuscript 'Schistosoma japonicum cathepsin B2 (SjCB2) facilitates parasite invasion through the skin' has been provisionally accepted for publication in PLOS Neglected Tropical Diseases.

Best regards,

Katja Fischer

Guest Editor

Michael Hsieh

Deputy Editor

---

## [Editor Report · Acceptance letter]

14 Oct 2020

Dear Professor Hu,

We are delighted to inform you that your manuscript, "Schistosoma japonicum cathepsin B2 (SjCB2) facilitates parasite invasion through the skin," has been formally accepted for publication in PLOS Neglected Tropical Diseases.

Best regards,

Shaden Kamhawi

co-Editor-in-Chief

Paul Brindley

co-Editor-in-Chief
